# Development of a Novel RT-qPCR Detecting Method of Covert Mortality Nodavirus (CMNV) for the National Proficiency Test in Molecular Detection

**DOI:** 10.3390/v14071475

**Published:** 2022-07-05

**Authors:** Wei Wang, Shuang Liu, Liang Yao, Jitao Xia, Tingting Xu, Chong Wang, Chen Li, Qingli Zhang

**Affiliations:** 1College of Fisheries and Life Science, Shanghai Ocean University, Shanghai 201306, China; 13173113773@163.com (W.W.); yaoliang2019@163.com (L.Y.); xiajitaook@126.com (J.X.); 2Yellow Sea Fisheries Research Institute, Chinese Academy of Fishery Sciences, Function Laboratory for Marine Fisheries Science and Food Production Processes, Qingdao National Laboratory for Marine Science and Technology, Key Laboratory of Maricultural Organism Disease Control, Ministry of Agriculture, Qingdao Key Laboratory of Mariculture Epidemiology and Biosecurity, Qingdao 266071, China; liushuang@ysfri.ac.cn (S.L.); xutingting83@163.com (T.X.); wangchongyilin@163.com (C.W.); lichen@ysfri.ac.cn (C.L.)

**Keywords:** TaqMan probe based reverse transcription quantitative PCR (TaqMan RT-qPCR), covert mortality nodavirus (CMNV), optimized method, rapid molecular detection, national proficiency test

## Abstract

Covert mortality nodavirus (CMNV), the pathogen of viral covert mortality disease (VCMD), has caused serious economic losses of shrimp aquaculture in Southeast Asian countries and China in the past decade. In view of that the rapid and accurate laboratory detection of CMNV plays a major role in the effective control of the spread of VCMD. The national proficiency test (NPT) for the detection of covert mortality nodavirus (CMNV) started in China from 2021. In this study, a novel TaqMan real-time reverse transcription quantitative PCR (RT-qPCR) detection method for CMNV with higher sensitivity than previous reports was established based on specific primers and probe designing from the conserved regions of the CMNV coat protein gene for using molecular detection of CMNV in NPT. The optimized RT-qPCR reaction program was determined as reverse transcription at 54.9 °C for 15 min and denaturation at 95 °C for 1 min, followed by 40 cycles including denaturation at 95 °C for 10 s, and annealing and extension at 54.9 °C for 25 s. The detection limit of the newly developed RT-qPCR method was determined to be as low as 2.15 copies of CMNV plasmids template per reaction, with the correlation coefficient (R^2^) at above 0.99. The new method showed no cross reaction with the six common aquatic animal pathogens and could be finished in one hour, which represents a rapid detection method that can save 50% detection time versus the previously reported assay. The CMNV TaqMan probe based RT-qPCR method developed in present study supplies a novel sensitive and specific tool for both the rapid diagnosing and quantitating of CMNV in NPT activities and in the farmed crustaceans, and will help practitioners in the aquaculture industry to prevent and control VCMD effectively.

## 1. Introduction

In 2014, a new nodavirus associated with infectious covert mortality disease (CMD) of shrimp was first identified in cultured *Penaeus vannamei*, and the virus was named as covert mortality nodavirus (CMNV) for being the pathogen of CMD [1,2]. The CMD was then renamed as viral covert mortality disease (VCMD) due to being caused by a viral pathogen [3]. CMNV is a non-enveloped, spherical virus with diameter about 32 nm [2]. It can infect the crustaceans, teleost and echinoderm, including *P. vannamei*, *P. monodon*, *P. chinensis*, *P. japonicus* [4], wild crabs [3,5] and fish [6,7,8,9], as well as sea cucumber [10]. CMNV infection was shown to cause cumulative mortality up to 80% in farming shrimp and cause the death of about 53% of infected zebrafish in an artificial infection experiment [6]. Recently, CMNV showed a high prevalence and wide distribution in the major shrimp cultured countries in Asia and Latin America [3,4,11,12].

Shrimp is one of the major species for the international trade of aquatic products and has significant benefits in socioeconomic terms [13]. In the past decades, outbreaks of diseases have largely restricted the development of global shrimp aquaculture, causing annual losses of tropical shrimp production of up to 40% [14,15]. Shrimps were reported to rely mainly on their innate immune system for immune protection, and lacking the specific immune system [3,16,17] makes it difficult to respond to the threat of virus epizootics in farming shrimp by using the strategy of developing and applying vaccines. Therefore, developing molecular biological detection methods with high specificity and sensitivity, the screening of CMNV as early, accurately and specifically as possible in breeding and farming practice and cutting off the transmission route of CMNV entering into the aquaculture system are considered the main technical solutions for the effective prevention and control of VCMD at present. Additionally, according to the International Epizootic Office (OIE) aquatic manual, infectious disease can be diagnosed by direct or indirect detection of infectious agents, and the molecular biological pathogen detection is most often used for laboratory diagnosis. Therefore, the molecular detection methods including nested PCR and real-time quantitative PCR have been considered to be the gold standard CMNV-specific tests to officially report a CMNV outbreak at present.

Under the National Aquatic Animal Disease Surveillance Program (NAADSP), national wide surveillance of major crustacean pathogens prevalence was implemented annually in China. To guarantee standardization of molecular detection of crustacean pathogens, participating laboratories of NAADSP were required to achieve certification from national proficiency test (NPT) by the fishery authority of China. In 2021, the nested PCR was chosen as the molecular detection method for CMNV in the NPT. According to the 2021 NPT assessment result, a certain proportion of participating laboratories encountered the problem of aerosol contamination of PCR amplification products in the proficiency test. Finding or developing new detection technologies for CMNV with low risk of contamination has become an urgent task at present. Several other molecular detection methods for CMNV have been reported in the past years, including real-time reverse transcription quantitative PCR [10,18] and the specific real-time reverse transcription loop-mediated isothermal amplification (RT-LAMP) [19]. These detection methods showed a few disadvantages in aspects of the respective diagnostic sensitivity, specificity and efficiency; thus, the method to detect CMNV more quickly, efficiently and accurately still needs further research and exploration.

Compared with conventional PCR, the detection and analysis process for fluorescence real-time quantitative PCR is completed automatically by the equipment in closed single tubes, and has the advantages of high automation and is an effective solution to the problem of PCR product contamination and quantitative detection of pathogen infection, and so real-time quantitative PCR is currently considered the most ideal method for pathogen detection in laboratories [20,21,22]. For supplying more suitable, sensitive and efficient detecting methods for CMNV in the NPT activities and in the monitoring pathogens of farmed crustaceans, we designed new primers and probes targeting the CMNV coat protein (CP) gene, and then established an improved one-step real-time Taqman based reverse transcription quantitative PCR detecting method (Taqman RT-qPCR) of CMNV in present study.

## 2. Materials and Methods

### 2.1. Sample Collection and RNA Extraction

A total of 300 live shrimp and prawn samples, including *P. vannamei* with body length of 3–8 cm and *Macrobrachium rosenbergii* with body length of 8–9 cm, were collected from ponds of shrimp farms in Shandong and Jiangsu provinces in China. Some diseased samples showed clinical symptoms of CMNV infection such as hepatopancreatic atrophy with color fading, empty stomach and guts, shell softening, both muscle whitening and necrosis at acute stage, etc. Cephalothoraxes and abdominal segment of the shrimp were cut into three parts along the central axis of the shrimp’s body and sampled. One part was preserved in 4% paraformaldehyde solution (PFA-PBS) (Sinopharm, Beijing, China) for in situ hybridization (ISH) assay. Another part was cut into strips (diameter approximately of 1 mm^3^) and fixed with TEM fixative (2% paraformaldehyde, 2.5% glutaraldehyde, 160 mM NaCl and 4 mM CaCl_2_ in 200 mM PBS, pH 7.2) as samples for electron microscopic analysis. The remaining tissues were minced, divided into two parts and preserved, respectively, in RNAlater RNA Stabilization Reagent (Qiagen GmbH, Hilden, Germany) and 95% ethanol for further molecular biological analysis. The total RNA was extracted from RNAstore-preserved tissues (30 mg) by using a commercial RNA Rapid Extraction Kit (Bio Teke, Beijing, China) according to the manufacturer’s instructions. The quality and concentration of the extracted RNA was measured by using Nanodrop 2000 (Thermo Scientific, Waltham, MA, USA). The extracted RNA was stored at −80 °C prior to use.

### 2.2. Primers and Probe

The coat protein (CP) gene in the RNA2 genome of CMNV (GenBank accession number MZ643944) was chosen as the targeted gene for designing primers. The primers and probe were designed by using primer design tools from IDT web (https://www.idtdna.com/pages/tools/primerquest?cUSS, accessed on 4 March 2022). The designed forward and reverse primers were CMN-CP-TaqIDT-F2 (5′-AACTACATCTGCACCCCATG-3′) and CMN-CP-TaqIDT-R2 (5′-TTGATGGTGTCGCTAGTCTTC-3′), respectively. The amplicon yielded by the primers was 144 bp in length. The TaqMan probe (5′-ATCCCTGCCGCTTAATGTGAGATCG-3′) was synthesized by Sangon (Shanghai, China), labeled with a 6-carboxyfluoresin (FAM) at the 5′ end and with a TAMRA^TM^ quencher (TAMRA) at the 3′ end, and purified by high-performance liquid chromatography (HPLC).

### 2.3. Preparation of Plasmid Standard and RNA Standard

The 144 bp CMNV amplicon amplified with primers CMN-CP-TaqIDT-F2 and CMN-CP-TaqIDT-R2 was cloned into the pMD18-T vector (TaKaRa, Dalian, China), and the recombinant plasmid sequence was then confirmed by gene sequencing (Sangon, Shanghai, China). The confirmed pMD18-CMNV recombinant plasmid was enriched by using the plasmid small quantity preparation kit (Omega, New York, NY, USA). The concentration of the recombinant plasmid was determined by the Nanodrop 2000 (Thermo Scientific, Waltham, MA, USA), and the copy number of target plasmid was calculated by using the tool from web (https://www.technologynetworks.com/tn/tools/copynumbercalculator, accessed on 19 April 2022). Tenfold dilution series solution of the recombinant plasmid was prepared and used as the plasmid standards of CMNV detection. RNA fragments containing the 144 bp sequence targeted CMNV CP gene was synthesized by TaKaRa (Dalian, China) and used as the RNA standards.

### 2.4. TaqMan RT-qPCR Assay and Procedure

The CMNV real-time RT-qPCR was performed in a BIORAD CFX96 Touch Real-Time PCR Detection System (BIORAD, Hercules, CA, USA) according to the instructions of Luna^®^ Universal Probe One-step RT-qPCR Kit (New England BioLabs, Ipswich, MA, USA). Optimization of reaction parameters and processes included reverse transcription at temperature of X °C (the X was set as 53 °C, 54.9 °C, 57.3 °C, 59.3 °C, 60.4 °C, 61 °C, respectively) for 15 min, followed by 40 cycles of denaturation at 94 °C for 10 s, and annealing and extension at temperature of X °C (the X was set as 51 °C, 51.7 °C, 53 °C, 54.9 °C, 57.3 °C, 59.3 °C, 60.4 °C, 61 °C, respectively) for 25 s. The 20 μL optimized reaction system included reaction components of reagent 1 Luna^®^ Universal Probe One-step Reaction Mix (2×) X μL (the X of reagent 1 was set as 7 μL, 8 μL, 9 μL, 10 μL, 11 μL, 12 μL, respectively), reaction components of reagent 2 Luna^®^ WarmStart RT Enzyme Mix X μL (the volume of reagent 3 was set as 0.4 μL, 0.6 μL, 0.8 μL, 1 μL, 1.2 μL, respectively), primers and probe X μL (the volume X of primers and probe were set as 0.1 μmol/L, 0.2 μmol/L, 0.3 μmol/L, 0.4 μmol/L, 0.5 μmol/L, 0.6 μmol/L, respectively). Finally, Data were analyzed with BIORAD CFX96 software (Version 6.0.14) to obtain the optimum reaction procedure. Data were analyzed through one-way analysis of variance (ANOVA) followed by an unpaired, single-tailed *t*-test. *p* value of <0.05 was considered to be statistically significant. The optimal temperature, time and concentration were determined by considering the minimum standard deviation (SD), minimum threshold cycle (*Ct*), economic cost and stability.

### 2.5. Analysis Specificity

The specificity of the CMNV TaqMan RT-qPCR was tested by using the nucleic acid of shrimp infected with common pathogens, which included infectious hypodermal and hematopoietic virus (IHHNV), shrimp hemocyte iridescent virus (SHIV), enterocytozoon hepatopenaei (EHP), white spot syndrome virus (WSSV), yellow head virus (YHV), *Vibrio* causing acute hepatopancreatic necrosis disease (*V*_AHPND_) and *Macrobrachium rosenbergii* nodavirus (MrNV). The RNA of *P. vannamei* infected with CMNV, the RNAs of healthy *P. vannamei*, and double distilled water were used as the positive, negative and blank controls, respectively. Each assay was repeated three times.

### 2.6. Standard Curve and Analysis Sensitivity

Standard curves were established using tenfold serial diluted solutions of pMD18-CMNV plasmids and RNA standard solutions as templates to test the sensitivity of TaqMan RT-qPCR, respectively. The standard curve was established from the measured *Ct* value (y) relative to the logarithmic concentration of the starting template (x) by BIORAD CFX96 Software (Version 6.0.14).

### 2.7. Repeatability Test

The inter- and intra-assay repeatability of the TaqMan RT-qPCR assay was tested by 10-fold dilutions of plasmid standards. The experiment of inter- and intra-assay repeatability were performed three times, respectively. All reactions with definite concentration were performed in triplicate. The repeatability of the TaqMan RT-qPCR assay was assessed using the coefficient of variation (CV), which is defined as the percentage of the standard deviation (SD) to the mean of *Ct* for each of the different pMD18-CMNV dilutions in this study. The analysis of variance (ANOVA) was performed by using SPSS program (Version 13, SPSS Inc., Chicago, IL, USA).

### 2.8. Clinical Implement Test

The TaqMan RT-qPCR assay previous reported by Li et al., (2018) has been adopted as the industry standard for CMNV testing in China. For comparing the newly developed assay’s performance, the newly developed TaqMan RT-qPCR assay and Li’s TaqMan RT-qPCR assay [18] were respectively used to test 300 clinical samples potentially infected with CMNV in the present study. The diagnostic sensitivity (DSe) and diagnostic specificity (DSp) of the new assay was determined according to the method recommended by OIE aquatic manual [23].

### 2.9. Probe Preparation and in Situ Hybridization (ISH)

The 244 bp CMNV RNA probes were prepared according to the protocols reported previously [5]. The clinical samples with different test results in the above-mentioned clinical tests were submitted for confirmation of CMNV by using ISH according to the procedures described previously [24]. The sections post ISH were counterstained using the Nuclear Fast Red solution (Solarbio, Beijing, China), then visualized under the Nikon Eclipse E80i microscope (Nikon Co., Tokyo, Japan), and finally imaged through the slide scanning system of Pannoramic MIDI (3DHISTECH Ltd., Budapest, Hungary).

## 3. Results

### 3.1. Optimization of TaqMan RT-qPCR Assay for CMNV

According to the principles of best TaqMan RT-qPCR process with the minimum SD, minimum *Ct*, economic cost and stability, the optimized result of reaction parameters are shown in Figure 1, and the optimized reaction procedure is shown as the following: 54.9 °C for 15 min, 95 °C for 1 min; 40 thermal cycling amplifications (95 °C for 10 s, 54.9 °C for 25 s), and the optimized reaction mixture contained 0.3 μM of each primer, 0.3 μM probe, 0.8 μL Luna WarmStart^®^ RT Enzyme Mix, 10 μL reagent 1 Luna^®^ Universal Probe One-step Reaction Mix (2×), 1 μL template, 6.4 μL RNA-free H_2_O.

### 3.2. Analysis Sensitivity and Standard Curve of TaqMan RT-qPCR Assay for CMNV

Using the 10-fold dilution series pMD18-CMNV plasmid standards of CMNV as the template, the test results revealed that the analysis sensitivity of the TaqMan RT-qPCR assay was reliable, and the amplification curve of each concentration gradient standard was reproducible and the fluorescence signal intensity of every reaction was steady (Figure 2A,B). The TaqMan RT-qPCR was capable to detect plasmid standards of CMNV as low as 2.15 copies per reaction. A linear relationship between the starting plasmid standards template concentration and *Ct* value was *Ct* = −3.157 log (starting quantity, Sq) + 38.567 (R^2^ = 0.998), within the starting template range of 2.15 × 10^9^ copies/μL to 2.15 × 10^0^ copies/μL.

Using the 10-fold dilution series RNA standards of CMNV as the template, the linear relationship between the starting RNA standards template concentration and *Ct* value was *Ct* = −3.243 log (starting quantity, Sq) + 38.932 (R^2^ = 0.997), within the starting template range of 1.36 × 10^9^ copies/μL to 1.36 × 10^0^ copies/μL (Figure 2C,D). The detection limit of the newly TaqMan RT-qPCR assay was 1.36 copies RNA standards of CMNV per reaction.

Additionally, the analysis sensitivity was also tested by mixing the CMNV RNA standard with the total RNA of shrimp tissue, and the results showed that a 1.36 copies/reaction detection limit can be achieved when using the total RNA of healthy shrimp tissues containing 1.36 copies CMNV standard RNA as template (Appendix A).

### 3.3. Analysis Specificity of TaqMan RT-qPCR Assay for CMNV

In the analysis specificity test for the newly developed TaqMan RT-qPCR assay, amplification curves were only obtained when the nucleic acids of CMNV were used as a template. There were no amplicons produced when using the nucleic acids of IHHNV, SHIV, EHP, WSSV, *V*_AHPND_, YHV or MrNV as the template (Figure 3). That is, the results indicated the TaqMan RT-qPCR assay is specific for CMNV in nucleic acid detection.

### 3.4. Repeatability of TaqMan RT-qPCR Assay for CMNV

The repeatability test for the newly developed CMNV TaqMan RT-qPCR assay showed that its intra- and inter-assay CV were less than 2.28% and 3.55%, respectively (Table 1), when using starting template concentrations ranging from 2.15 × 10^0^ to 7.63 × 10^9^ copies per reaction. The results of the variance analysis showed that the *p* values of the inter-assay variation for each concentration of the diluted plasmid were >0.05, which indicated that the mean inter-assay *Ct* differences were not significant (Table 2).

### 3.5. Clinical Application of TaqMan RT-qPCR Assay for CMNV

The compare of the newly developed CMNV TaqMan RT-qPCR assay and another assay reported previously were conducted by using a total of 300 clinical samples [18]. The test results obtained by using Li’s previously reported method showed 79 of the 300 samples were CMNV-positive. The test results obtained by using the newly developed CMNV TaqMan RT-qPCR assay indicated 76 of 79 samples determined to be CMNV positive by Li’s previously reported method gave positive results; meanwhile, 12 samples out of 221 samples determined to be CMNV negative by Li’s method were detected to be CMNV positive in the test using the newly developed TaqMan RT-qPCR assay. Therefore, the DSe and DSp values for the newly developed TaqMan RT-qPCR method compared with Li’s previously reported method were 96.20% and 94.57%, respectively (Table 3). Meanwhile, the newly developed method showed quite good repeatability in the test for the clinical samples (Appendix A). 

## 4. Discussion

Several histopathological, ultra-histological and molecular methods for detecting CMNV have been reported over the past years. As traditional pathogen detection techniques, the histopathology and traditional PCR methods for CMNV diagnostics take a relatively long time to obtain results [2,11]. Although CMNV RT-LAMP assay is a rapid method and can be used in the field, its sensitivity is low and cannot be used for quantitative of low-dose CMNV samples [19]. In this study, we developed a new TaqMan RT-qPCR assay for CMNV detection and quantitation with much higher specificity and sensitivity than the methods previously reported.

Previous studies have found that the CMNV RdRp gene sequences of different isolates differ from the original CMNV RdRp gene sequences by 0% to 4% [3,9]. Theoretically, RdRp lacks proofreading activity, which leads to a high mutation rate in the RNA virus genome [25,26]. The primers of the molecular detection method should be selected within the conserved region of the target gene as far as possible without non-specific binding with the corresponding gene of the related species [27,28]. Hereby, we designed a set of TaqMan RT-qPCR primers and probes based on CP protein of CMNV for the development of new CMNV sensitive detection method. The newly designed primers and probes targeted the 144 bp region of the CMNV CP gene (Figure 4A), and have no cross reaction with the other six pathogens, except CMNV, which indicated that the method had strong specificity for detecting CMNV. 

We compared the TaqMan RT-qPCR assay established in this study with the assay developed by Li (2018), using the two methods to detect 300 clinical shrimp samples from different geographical origins. This comparison result revealed a discrepancy in 15 samples, and 12 out of the fifteen samples were CMNV negative in tests by using Li’s assay (2018) but were CMNV positive in tests by using the newly developed assay. The sequencing and alignment of CMNV RdRp genes from the discrepant *P. vannamei* above-mentioned samples revealed that single-base mutation occurred at 2965 nt of the RdRp gene from the sample that was CMNV positive in the newly developed method detection and CMNV negative in Li’s previously reported method detection (Figure 4B), and the single-base mutation was located at the 13th nucleic acid bases of the reverse primer of Li’s method (2018). Thus, it is likely that the single-base mutation in the CMNV variant isolates from the 12 samples was responsible for the detection failure using the Li’s method (2018). Further, the 12 discrepant shrimp samples that were CMNV negative determined by the Li’s assay (2018) but positive determined by this newly assay were proved to be infected by CMNV in the ISH assay (Figure 4C). The other three out of the 15 samples, which were determined to be CMNV positive by Li’s assay (2018) but to be CMNV negative by the newly developed TaqMan RT-qPCR, were eventually identified to be mild infection with CMNV by analysis of ISH. The authors tried to clone the CMNV target fragment of the RdRp gene of these three samples by CMNV RT-nPCR, but the RT-nPCR assay of these three samples generated a negative result. The failure of the RT-nPCR assay might cause by the low number of CMNV viral copies in these samples.

In all the reported CMNV test methods, CMNV TaqMan RT-qPCR assays reported previously by Pooljun (2016) and Li’s (2018) represented the typical method of molecular testing of CMNV, and have been recommended by the disease card of CMNV infection issued by Network of Aquaculture Centres in Asia–Pacific (NACA). The total reverse transcription and amplification time of the optimized CMNV TaqMan RT-qPCR assay in the present study is about one hour, which is about 50% shorter than the time consumed by TaqMan RT-qPCR assay for CMNV developed previously by Pooljun (2016), and is about 20% shorter than the time consuming by Li’s assay (2018). Additionally, the tissue RNA preparation time is usually 1 h, and the total test time is only about 2 h when combining the newly developed method in the present study. The optimized reactions in this study, including both of the reverse transcription and the nucleic acid target sequence amplification, can be efficiently completed in a relatively short time, and the change of the fluorescence signal of the probe is convenient to monitor the whole reaction process. Overall, the newly developed CMNV TaqMan RT-qPCR method is more time saving and efficient than the methods previously reported.

The sensitivity test revealed that the TaqMan RT-qPCR assay could detect CMNV RNA as low as 1.36 copies per reaction, which is more sensitive than those methods that developed for CMNV detection previously, including the TaqMan RT-qPCR (5.7 copies per reaction in Li’s 2018 report), and the LAMP methods (27 copies per reaction in Zhang’s 2017 report). In the present study, both of the 10-fold series dilution solutions of both CMNV plasmid and RNA standard were used as templates for constructing the standard curve according to *Ct* value to meet the future potential quantified need for using the cDNA and RNA templates. Based on the standard curve, the equations were calculated using regression analysis comparing the *Ct* value to the standard copy number. In the range of 10^9^–10^0^ RNA standards and 10^9^–10^0^ plasmid copies, the correlation coefficients were high (R^2^ = 0.997, R^2^ = 0.998), which indicates that the newly developed TaqMan RT-qPCR assay is an ideal quantitation tool of CMNV. In addition, the TaqMan RT-qPCR is conducted in closed tubes, and will effectively avoid the problem of product contamination caused by nested PCR.

In conclusion, this paper reports on a new TaqMan RT-qPCR assay for the quantitative detection of CMNV. This assay has the advantage of being rapid, sensitive and specific. The newly TaqMan RT-qPCR established in this study provides a novel and efficient method for CMNV detection, in both the national performance test and in the shrimp pathogen surveillance.

## Figures and Tables

**Figure 1 viruses-14-01475-f001:**
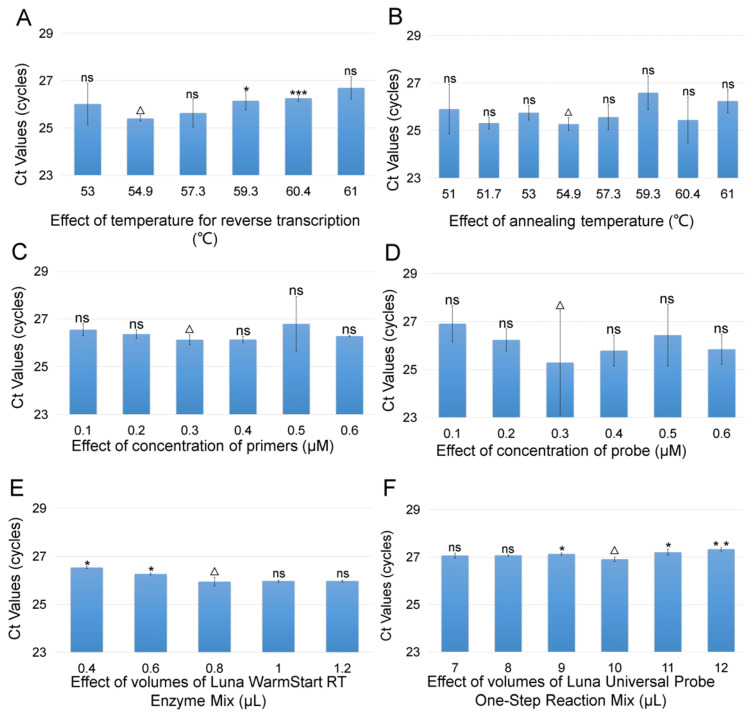
Optimization of the TaqMan RT-qPCR reaction for detection of the coat protein gene of covert mortality nodavirus (CMNV). (**A**) Effect of temperature for reverse transcription on the TaqMan RT-qPCR reaction. The columns from left to right represent 53 °C, 54.9 °C, 57.3 °C, 59.3 °C, 60.4 °C and 61 °C, respectively. (**B**) Effect of annealing temperature on the TaqMan RT-qPCR reaction. The columns from left to right represent 51 °C, 51.7 °C, 53 °C, 54.9 °C, 57.3 °C, 59.3 °C, 60.4 °C and 61 °C, respectively. (**C**) Effect of concentration of primers on the TaqMan RT-qPCR reaction. The columns from left to right represent 0.1 μmol/L, 0.2 μmol/L, 0.3 μmol/L, 0.4 μmol/L, 0.5 μmol/L and 0.6 μmol/L, respectively. (**D**) Effect of concentration of probe on the TaqMan RT-qPCR reaction. The columns from left to right represent 0.1 μmol/L, 0.2 μmol/L, 0.3 μmol/L, 0.4 μmol/L, 0.5 μmol/L, 0.6 μmol/L, respectively. (**E**) Effect of volumes of Luna^®^ WarmStart RT Enzyme Mix on the TaqMan RT-qPCR reaction. The columns from left to right represent 0.4 μL, 0.6 μL,0.8 μL,1 μL and 1.2 μL, respectively. (**F**) Effect of volumes of Luna^®^ Universal Probe One-Step Reaction Mix (2×) on the TaqMan RT-qPCR reaction. The columns from left to right represent 7 μL, 8 μL, 9 μL, 10 μL, 11 μL and 12 μL, respectively. ∆: The best result after optimization. Differences of each group are marked in the figure. ***: *p* ≤ 0.001, indicating the very significant statistical difference; **: *p* ≤ 0.01, indicating the significant statistical difference; *: *p* ≤ 0.05, indicating the statistical difference; ns: *p* > 0.05, indicating no statistical difference.

**Figure 2 viruses-14-01475-f002:**
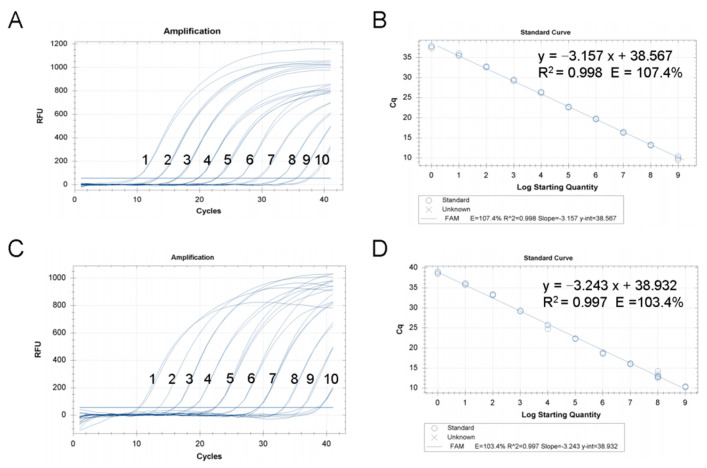
The standard curve of the newly developed covert mortality nodavirus (CMNV) TaqMan RT-qPCR assay. (**A**) The amplification plots of TaqMan RT-qPCR using serial 10-fold dilutions of the plasmids as templates. 1–10: 2.15 × 10^9^ to 2.15 × 10^0^. (**B**) Standard curve and standard curve equation for the CMNV TaqMan RT-qPCR assay using plasmids templates. (**C**) The amplification plots of TaqMan RT-qPCR using serial 10-fold dilutions of the standard RNA templates. 1–10: 1.36 × 10^9^ to 1.36 × 10^0^. (**D**) Standard curve and standard curve equation for the CMNV TaqMan RT-qPCR assay using RNA templates.

**Figure 3 viruses-14-01475-f003:**
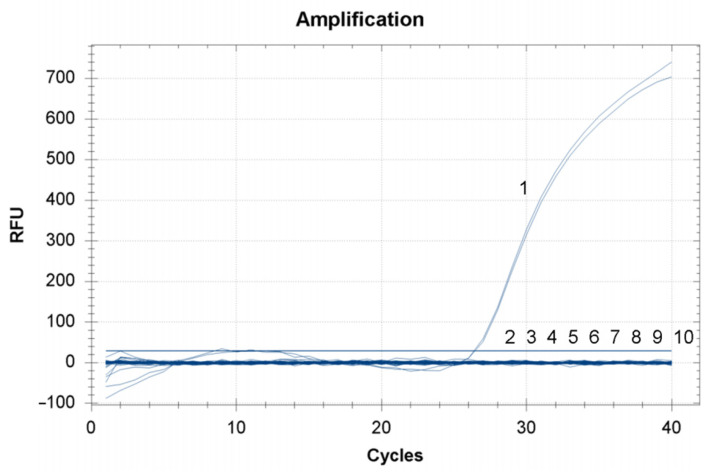
Analytical specificity test for the newly developed covert mortality nodavirus (CMNV) TaqMan RT-qPCR assay. 1. CMNV, 2. IHHNV, 3. SHIV, 4. EHP, 5. WSSV. 6. *Vp*_AHPND_, 7. YHV, 8. MrNV, 9. Negative control, 10. Blank control.

**Figure 4 viruses-14-01475-f004:**
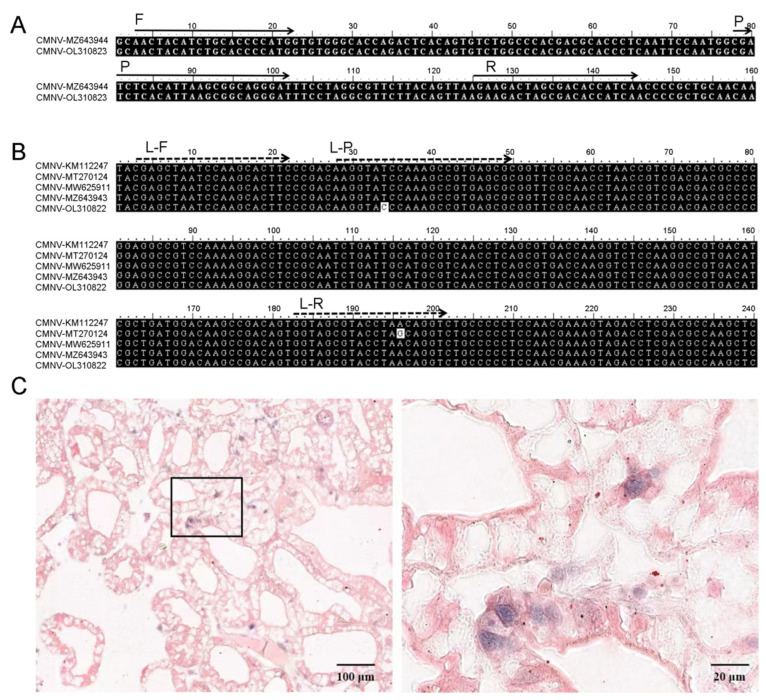
Analysis of the influence of single-base mutation in different covert mortality nodavirus (CMNV) isolate genome sequence on detection methods, and in situ hybridization (ISH) verification of CMNV infection in the discrepant sample detected by new and old methods. (**A**) Multiple sequence alignment of the targeted RdRp genes from different CMNV isolates, and the locates of primer and probe of Li’s assay (2018). GenBank accession number of the original CMNV isolates’ RdRp gene was KM112247 which was used to design TaqMan RT-qPCR primers and probes in previous Li’s assay (2018); GenBank accession number for CMNV RdRp genes from discrepant samples detected by new method and Li’s assay (2018) were MT270124, MW625911, MZ643943 and OL310822, respectively. Arrows indicate nucleotide sequences of primers and probes. L-F: forward primer of Li’s assay (2018); L-R: reverse primer of Li’s assay (2018); L-P: probe of Li’s assay (2018). (**B**) Multiple sequence alignment of the targeted CP genes from different CMNV isolates, and the locates of primers and probes of the newly developed TaqMan RT-qPCR assay. F: forward primer; R: reverse primer; P: probe. (**C**) ISH verification of CMNV infection in the discrepant sample detected by new developed assay and Li’s assay (2018). The sample was detected to be CMNV positive by newly developed assay, while it was detected to be CMNV negative by Li’s assay (2018). The tissue of Figure 4C was from *Penaeus vannamei* sample and the shrimp showed mild CMNV infection symptoms, such as softening shell and pale hepatopancreas. Note the light purple hybridization signal of the CMNV RNA probe was observed at sites locating hepatopancreatic epithelial cells in the ISH sections. The photo on the right is an enlargement of the boxed area in the picture on the left.

**Table 1 viruses-14-01475-t001:** Intra-assay and inter-assay variability of the newly developed TaqMan RT-qPCR assay.

Dilution of Plasmid	Intra-Assay *Ct*			INTER-Assay *Ct*		
(Copies/Reaction)	Mean	SD	CV (%)	Mean	SD	CV (%)
2.15 × 10^9^	10.34	0.11	1.07	9.98	0.27	2.72
2.15 × 10^8^	12.87	0.07	0.56	13.25	0.30	2.23
2.15 × 10^7^	16.08	0.13	0.81	16.26	0.18	1.09
2.15 × 10^6^	18.71	0.18	0.98	19.34	0.69	3.55
2.15 × 10^5^	22.35	0.05	0.21	22.44	0.28	1.26
2.15 × 10^4^	25.42	0.58	2.27	25.61	0.81	3.17
2.15 × 10^3^	29.22	0.12	0.43	29.48	0.37	1.24
2.15 × 10^2^	33.29	0.13	0.39	32.87	0.36	1.11
2.15 × 10^1^	35.95	0.24	0.67	35.68	0.56	1.57
2.15 × 10^0^	38.75	0.28	0.73	38.27	0.49	1.28

**Table 2 viruses-14-01475-t002:** Analysis of variance for the inter-assay variability of the newly developed TaqMan RT-qPCR assay.

Dilution of Plasmid (Copies/Reaction)	Inter-Assay *Ct*	
	F Value	*p* Value
2.15 × 10^9^	4.409	0.310
2.15 × 10^8^	0.020	0.213
2.15 × 10^7^	2.896	0.109
2.15 × 10^6^	3.423	0.460
2.15 × 10^5^	3.347	0.297
2.15 × 10^4^	5.614	0.435
2.15 × 10^3^	2.523	0.291
2.15 × 10^2^	0.048	0.777
2.15 × 10^1^	0.378	0.460
2.15 × 10^0^	8.520	0.171

**Table 3 viruses-14-01475-t003:** The comparison of detection results of the newly developed TaqMan RT-qPCR and Li’ s (2018) TaqMan RT-qPCR.

	Number of Reference Samples	
	Positive Samples, 79	Negative Samples, 221
Detected positive	76	12
Detected negative	3	209
Calculation	DSe = 76/79 × 100% = 96.20%	DSp = 209/221 × 100% = 94.57%

## Data Availability

Data available on request from the authors.

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
