# Peer review of "Development of a Novel RT-qPCR Detecting Method of Covert Mortality Nodavirus (CMNV) for the National Proficiency Test in Molecular Detection"

_viruses, 2022, doi:10.3390/v14071475_

Round 1

Reviewer 1 Report

Real-time quantitative PCR has been widely used in recent years for disease diagnostic. Here, an important viral pathogen of shrimp was under study. The goal of this work was to develop a TaqMan qPCR assay for the shrimp nodavirus CMNV more efficient than previously reported PCR detection methods. That should mean a more sensitive and less time-consuming RT-qPCR test for CMNV. The qPCR protocol reported in here targets the virus CP gene that is presumably more conserved than the RdRp gene targeted in earlier works.

In general, all the expected experimental evidence seems to be in place: optimization, standard curves, specificity, repeatability, and the comparison with a previously reported qPCR assay for CMNV-infected shrimp. The RT-qPCR was put to the test on samples taken from diseased fish and seemed to perform better than the previous methods.

Minor issues

Introduction: it is not clear what is the gold standard CMNV-specific test to officially report a CMNV outbreak.

Methods (2.8) Please provide a more detailed description of the clinical samples.

Figure 1. I believe data will be better presented in one separated graph (in line format) per each one of the parameters, with the X-axis showing the different values of the parameter.

Sensitivity of the method: is it correct that the sensitivity of the assay is equal with plasmid DNA and with RNA (1.36 copies/reaction)? Also regarding sensitivity: could the 1.36 copies/reaction detection limit be achieved by spiking shrimp tissues with synthetic RNA?

Specificity of the method: I have a slight concern on the qPCR assay not being tested for other shrimp nodaviruses (PvNV, MrNV). TaqMan qPCR tests should be specific for the targeted sequence. But still, it would have been a nice piece of data to have tested at least one other crustacean nodavirus.

Repeatability: was the repeatability checked with field samples (like those in table 3)? Would you get the same number of positive and negative cases if you repeat the assay?

In lines 334- 336, Regarding the time duration of the qPCR assay: Did the authors take into account the time to do the RNA extractions? How many hours would the total procedure take?

Additionally, I don´t quite understand why the authors compare to the time period for the method of Pooljun (2016) while in the rest of the manuscript they were comparing their qPCR with the one reported in Li et al. (2018).

Reviewer 2 Report

  1. Please complete the introduction with OIE data (link - 

    https://www.oie.int/fileadmin/Home/fr/Our_scientific_expertise/reflabreports/2018/report_635_2018_Infectious_hypodermal_and_haematopoietic_CHINA_(PEOPLES_REP__OF).pdf

    and the publication from 2018 (link - https://pubmed.ncbi.nlm.nih.gov/30308216/
  2. Please provide the number of samples taken from each species of shrimp, not the total number
  3. Please provide information on whether the specimens from which the samples were taken showed any clinical symptoms of CMNV.
  4. Provide information if the cultures sampled had a CMNV history
  5. Please provide in the description of Figure 4C what species the sample was from and whether the subject showed clinical symptoms of CMNV.
